# A Pneumatic Soft Exoskeleton System Based on Segmented Composite Proprioceptive Bending Actuators for Hand Rehabilitation

**DOI:** 10.3390/biomimetics9100638

**Published:** 2024-10-18

**Authors:** Kai Li, Daohui Zhang, Yaqi Chu, Xingang Zhao, Shuheng Ren, Xudong Hou

**Affiliations:** 1The State Key Laboratory of Robotics, Shenyang Institute of Automation, Chinese Academy of Sciences, Shenyang 110016, China; likai@sia.cn (K.L.); zhaoxingang@sia.cn (X.Z.); renshuheng@sia.cn (S.R.); houxudong@sia.cn (X.H.); 2The Institutes for Robotics and Intelligent Manufacturing, Chinese Academy of Sciences, Shenyang 110169, China; 3University of Chinese Academy of Sciences, Beijing 100049, China

**Keywords:** predicted joint stiffness, extension angle, segmented composite proprioceptive bending actuators (SCPBAs), bending resistance, soft hand exoskeleton system

## Abstract

Soft pneumatic actuators/robotics have received significant interest in the medical and health fields, due to their intrinsic elasticity and simple control strategies for enabling desired interactions. However, current soft hand pneumatic exoskeletons often exhibit uniform deformation, mismatch the profile of the interacting objects, and seldom quantify the assistive effects during activities of daily life (ADL), such as extension angle and predicted joint stiffness. The lack of quantification poses challenges to the effective and sustainable advancement of rehabilitation technology. This paper introduces the design, modeling, and testing of pneumatic bioinspired segmented composite proprioceptive bending actuators (SCPBAs) for hand rehabilitation in ADL tasks. Inspired by human finger anatomy, the actuator’s soft-joint–rigid-bone segmented structure provides a superior fit compared to continuous structures in traditional fiber-reinforced actuators (FRAs). A quasi-static model is established to predict the bending angles based on geometric parameters. Quantitative evaluations of predicted joint stiffness and extension angle utilizing proprioceptive bending are performed. Additionally, a soft under-actuated hand exoskeleton equipped with SCPBAs demonstrates their potential in ADL rehabilitation scenarios.

## 1. Introduction

Stroke is a leading cause of disability worldwide, with over 13.7 million new patients annually. Stroke survivors often suffer from motor dysfunction and require external assistance for basic activities of daily life (ADL), which is crucial for maintaining quality of life [1]. The absence of independence in ADL diminishes their quality of life and reduces life satisfaction. Unfortunately, approximately 65% of stroke survivors face hand impairments that significantly impede their ability to perform ADL tasks [2]. Thus, various mechanical exoskeletons have been developed to aid in hand rehabilitation and restore survivors’ independence [3,4].

Traditional hand rehabilitation systems typically rely on rigid links and electric motors [5], which have the distinctive characteristics of cumbersome support, easily causing secondary damage to wearers, and interfering with normal motion in performing ADL tasks. Thus, some wearable pneumatic hand rehabilitation exoskeletons made of soft/elastic material came into being, possessing the advantages of being light, with a high power-to-weight ratio, of low cost, and being comfortable [6]. Polygerinos et al. developed an elastomer-based robotic glove to augment hand rehabilitation at home for individuals with function grasp pathologies, which was able to carry out gross and precise grasping [7]. Yap et al. presented a fully fabric-based bidirectional soft glove design to assist hand-impaired patients, which can realize active finger flexion and extension for training [8]. Correia et al. created a textile-based soft robotic glove controlled by the user with a button and then tested it in thirteen tetraplegic patients using the Jebsen–Taylor hand function test, which showed the effectiveness of this system [9]. Li et al. proposed a variable stiffness pneumatic actuator coupled with a multi-stage articulated steel layer and verified the extension effect of the actuator over a clenched artificial hand with damper force [10].

Various pneumatic soft gloves have been conducted in typical hand rehabilitation scenarios to validate their performance [9,11,12,13,14], but few exoskeleton systems can directly quantitatively characterize the assistant effect after wear in ADL tasks, such as extension angle and predicted joint stiffness. The existing hand function scale of the modified Ashworth scale (MAS) is subjective and affected by the doctor’s experience, which is not conducive to evaluation [15,16]. Shi et al. replicated a standalone desktop device to measure the passive joint moment and angle for the MCP joint in the index finger [17]. However, the bulky size of the test bench limits the potential to characterize the other joints and take advantage of portability. Heung et al. introduced a soft-composite rehabilitation actuator that objectively reflects the condition of post-stroke, which was validated by simulations and experiments [18,19]. Matsunaga et al. combined information from a joint modular soft actuator and a marker-less hand joint position acquisition device for finger joint stiffness estimation in a tele-rehabilitation environment [20]. However, it has limited potential for wide applications because an external camera is required to supervise the bending process during operation. Matsunaga et al. proposed artificial neural network (ANN)-based models to simultaneously estimate the stiffness of the distal interphalangeal (DIP) joint, the proximal interphalangeal (PIP) joint, and the metacarpophalangeal (MCP) joint for index fingers [21]. Still, it is challenging to prove the validity of the stiffness estimation methods and requires a lot of training data from dummy fingers using machine learning. Lai et al. focused on a soft pneumatic glove with honeycomb pneumatic actuators (HPAs) for assisting ADL tasks and developed two customized finger-force products for quantifying enhanced finger function in patients [22]. The primary function of these systems is almost always performance evaluation, and they cannot simultaneously be employed in rehabilitation ADL tasks.

For different pneumatic soft robotic systems designed for hand rehabilitation/assistance, as listed in Table A3, various rehabilitation training modes have been explored for an integrated “human–soft exoskeleton” system in ADL tasks. Sui et al. engineered a soft-packaged rehabilitation glove with tight integration of sensing, actuation, human–machine interference, and a closed-loop algorithm to regain fine motor skills in hand rehabilitation [12]. Zhou et al. adopted a state machine controller based on signals from integrated sensors to detect users’ intuition in hand–object interactions, inflating the corresponding actuators at maximum operation pressure [23]. Chen et al. provided a tendon-driven soft hand exoskeleton with a hybrid configuration based on a graphical user interface (GUI) open-loop control strategy, in which the training paradigm parameters are directly predefined by the kinematic model [24]. Chen et al. also accomplished dexterous hand/forearm manipulation with the assistance of a foot-controlled interface by utilizing movements of the unaffected foot as command signals, which is an open-loop limb mapping control scheme [25]. Tang et al. proposed a model-based online learning adaptive control algorithm for a wearable soft robotic glove by predefined trajectory, taking its interaction with the human fingers into account [26].

In this study, we propose a pneumatic soft hand exoskeleton system based on SCPBAs that incorporates a fiber-elastic body and stiffness-compensating layer-integrated bending sensors to realize predicted finger joint stiffness, reverse stretching motion, and rehabilitation training in ADL tasks simultaneously. First, the bioinspired principle behind and fabrication of the SCPBAs are presented. A flexible-joint–rigid-bone segmented configuration inspired by the anatomy of the human finger is put forward for the proposed SCPBAs. Next, the analytical models of the SCPBAs for both free space and constrained space are presented. The stiffness-compensating layer integrated proprioceptive sensors attached to the bottom of the SCPBAs can monitor the bending deformation and reversely extend for hand dysfunction, such as a clamp-shape hand. A reorganized analytical model that establishes the relationship between the inflated pressure, bending angle, and joint stiffness is also applied to evaluate finger joint stiffness in an integrated finger–actuator condition. Performance characterizations using the dummy/mannequin finger and task-oriented training strategies involving one healthy subject were conducted. The soft hand exoskeleton realized the bending deformation tracking as well as rehabilitation training in ADL tasks, such as tripod pinch and tip pinch.

## 2. Design and Fabrication of the SCPBAs

In this section, based on our previous work [27], the proposed SCPBAs inspired by the anatomy of the human finger are formulated first. Then, the improved modular molding method is used to prototype the actuators, which is also called secondary post-processing.

### 2.1. The Bio-Inspiration and Implementation of the SCPBAs

As in the anatomical diagram of the hand/finger in Figure 1a [27,28], the structure of the finger consists of phalanx/bone, joint, skin, muscle/tendon, and ligament. First, from the perspective of hand motor function, the phalanges have been linked with three joints: the MCP joint, the PIP joint, and the DIP joint. The tendon transforms the actuation force from the muscle to the corresponding finger joint. Second, from the ontological sensing function, the skin acts as a soft cushion, as well as the proprioceptor, which can guarantee non-destructive interaction and also provide perception information.

Considering the two aforementioned aspects of the finger/digit, the proposed bionic SCPBAs can greatly conform to the profiles of the finger and also provide motion information in real time. The principles of its biomimetic design are as follows.

(1) First, from the perspective of structural motion biomimetics, distinct from traditional fiber-reinforced actuators (FRAs), the SCPBA assembly incorporates rigid rings not only simply mimicking the soft-joint–rigid-bone anatomy of the fingers but also provides multi-joint/segmented bending morphing, which can enhance the conformability interaction of the finger joint. In other words, the proposed SCPBAs can promote segmented bending on the flexible joint section and keep the matrix from deforming in the rigid phalangeal sections where bending is not desired during the whole actuated process. This secondary post-processing method, called mechanical programming, relies on the utilized materials’ mechanical properties.

(2) Next, to realize the sensing function of the abovementioned fingers, a stiffness-compensating layer with a sensing function element is attached to the bottom surface of the SCPBAs, which can simultaneously enhance body stiffness and monitor bending deformation.

As shown in Figure 1b, the pneumatic SCPBAs incorporate a fiber-reinforced semi-obround elastomer embedded with rigid rings to mimic the soft-joint–rigid-bone anatomy of human fingers. The semi-obround matrix is surrounded by a network of fibers (fiber-reinforced layer), several rigid segments (the bottom strain-limiting layer, the stiffness-compensating layer, and rigid rings), and an outer elastomeric skin (outer wrappage). However, the DIP joint contributes only 15% of the function grip. Thus, the proposed SCPBAs for a wearable exoskeleton are double-segmented and lack covering of the DIP joint.

### 2.2. Actuator Fabrication

The proposed SCPBAs have the advantages of scalable fabrication, personalized customization, and low cost. The SCPBAs consist of a semi-obround matrix surrounded by a network of fibers (fiber-reinforced layer), several rigid segments (the bottom strain-limiting layer, the stiffness-compensating layer, and rigid rings), and an outer elastomeric skin (outer wrappage).

The main fabrication and assembly of the bioinspired SCPBAs are detailed in our previous study [27]. The inner elastomeric tube (2 mm thickness) is made of silicone elastomer (Dragon Skin 30, Smooth on Inc., Macungie, the U.S.) (Figure 2A,B). A single polyethylene thread (0.165 mm thickness, Yunshangpiao Co., Ltd, Jinhua, China.) is wound around the inner tube, forming the fiber-reinforced layer to provide radial constraint during the inner tube inflation, and a flexible but inextensible strain-limiting layer is attached to the bottom of the fiber-reinforced layer to construct the flexion motion. The strain-limiting layer is a piece of fabric strip 0.2 mm thick (Figure 2C). The outer skin silicone elastomer, 0.5 mm thick (Ecoflex 00-30, Smooth on Inc., Macungie, PA, USA), with a lower Young’s modulus than the inner tube fixes all components and provides a cosmetic appearance (Figure 2D). Three rigid rings, made of resin by 3D printing with mounting slots, are attached to the elastomeric body to keep these sections straight upon pressurization (Figure 2G). The stiffness-compensating layer with flexible bending sensors (FlexSensor 2.2/4.5, Spectrasymbol) is placed in the mounting slots (Figure 2H). Inspired by the anatomy of human fingers, we designed the index, middle, ring, and little fingers with three rigid segments and the thumb with two rigid segments, as shown in Figure 2E,F,I.

## 3. Mathematical Modeling of the SCPBAs

In this section, we explicitly formulate the relationship between input pressure, bending angle, and joint stiffness to illustrate the behavior of the SCPBAs [18,19]. Firstly, the bending model with input pressure was developed in free space, allowing the motion characteristics of the actuators under different thicknesses of the stiffness-compensating layer. Next, the wearer’s hand condition, such as finger spasticity or hypertonia, which is prevalent in stroke survivors, is taken into consideration. Three quantitative indexes are used to describe the SCPBAs specifically.

Evaluation Index 1: Bending Resistance BREvaluation Index 2: Extension Angle (degree) *θ*_0_ − *θ*Evaluation Index 3: Stiffness Evaluation (Nm/rad) K*_J_*

### 3.1. Bending Model in Free Space

In the free space shown in Figure 3, the bending torque equilibrium equation of the SCPBAs is composed of the bending moment induced by the inflation pressure, the elastomeric-based body, and the bottom stiffness-compensating layer, thus:(1)$$M_{P}=M_{b}+M_{L}$$

*M_P_* is the bending moment induced by the input pressure *P_in_*, comprising the bending moment of the hemi-circular shape *M_HC_* and rectangle shape *M_R_*. The geometric parameters and their values are shown in Table A1. The sum bending moment of the applied pressure can be described as follows:(2)$$\begin{array}{l} M_{P}=M_{R}+M_{HC}\\ =\int_{0}^{b}wP_{in}\left(y+a+\frac{t}{2}\right)dy+\int_{0}^{r}2\sqrt{r^{2}-y^{2}}P_{in}\left(y+a+b+\frac{t}{2}\right)dy\\ =\left[w\left(ab+\frac{bt}{2}+\frac{b^{2}}{2}\right)+\frac{\pi r^{2}}{2}\left(a+b+\frac{t}{2}\right)+\frac{2r^{3}}{3}\right]P_{in} \end{array}$$

*M_b_* is the bending moment composed of the elastic bending torque in the bottom part *M_bottom_*, side part *M_side_,* and top part *M_top_*, i.e., *M_b_* = *M_top_* + *M_side_ + M_bottom_*. The proposed actuator body is fabricated from silicone rubber (Dragonskin 30, Smooth on Inc.), which can be molded as an Ogden first-order hyper-elastic model. Its strain energy is given by the following:(3)$$W\left(\lambda_{1},\lambda_{2},\lambda_{3}\right)=\frac{\mu_{1}}{\alpha_{1}}\left({\lambda_{1}}^{\alpha_{1}}+{\lambda_{2}}^{\alpha_{1}}+{\lambda_{3}}^{\alpha_{1}}-3\right)$$
where the material coefficient *α*_1_ is the strain-hardening exponent, *µ* is the initial shear modulus of the material, and $\mu_{1}=\frac{2\mu }{\alpha_{1}}$. The estimated values of the coefficients *µ*_1_ = 75,449 kPa and *α*_1_ = 5.836 can be acquired from a uniaxial compression test. The principal nominal stresses *s_i_* can be obtained as $s_{i}=\mu_{1}\left({\lambda_{i}}^{\alpha_{1}}-{\lambda_{i}}^{-\alpha_{1}}\right)$ for *i* = 1, 2, 3. Coefficients *λ*_1_, *λ*_2_, and *λ*_3_ denote the elongation in the axial, circumferential, and radial directions, respectively. The circumferential direction is wrapped around the non-stretchable fibers, which can induce *λ*_2_ = 1.

*Approximation* For the sake of convenience, we use the Maclaurin series for stress *s* to acquire the numerical representation between the input pressure and bending angle output, i.e., $s=\mu_{1}\left(\lambda^{\alpha_{1}}-\lambda^{-\alpha_{1}}\right)\approx \frac{2\mu_{1}\alpha_{1}\theta }{L}y$. Then, the corresponding bending torques are depicted (where *dτ* is the differential thickness element of the chamber and *dφ* is the differential angle element around the center of semicircle O):(4)$$\begin{array}{l} M_{top}=2\int_{0}^{a}\left(\int_{0}^{\frac{\pi }{2}}s_{top}\left(\left(r+\tau \right)\left(\left(r+\tau \right)\sin\phi +\left(a+b+\frac{t}{2}\right)\right)\right)d\phi \right)d\tau \\ \approx \theta \frac{\mu_{1}\alpha_{1}}{L}\left[\pi {\left(a+b+\frac{t}{2}\right)}^{2}\left(a^{2}+2ar\right)+\frac{4\pi }{3}\left(a+b+\frac{t}{2}\right)\left[{\left(a+r\right)}^{3}-r^{3}\right]+\frac{\pi }{4}\left[{\left(a+r\right)}^{4}-r^{4}\right]\right] \end{array}$$
(5)$$M_{side}=2\int_{\frac{t}{2}}^{\frac{t}{2}+a+b}ays_{side}dy\approx 2\int_{\frac{t}{2}}^{\frac{t}{2}+a+b}a\frac{2\mu_{1}\alpha_{1}\theta }{L}y^{2}dy=\theta \frac{4\mu_{1}\alpha_{1}a}{3L}\left[{\left(\frac{t}{2}+a+b\right)}^{3}-{\left(\frac{t}{2}\right)}^{3}\right]$$
(6)$$M_{bottom}=\int_{\frac{t}{2}}^{\frac{t}{2}+a}wys_{bottom}dy\approx \int_{\frac{t}{2}}^{\frac{t}{2}+a}w\frac{2\mu_{1}\alpha_{1}\theta }{L}y^{2}dy=\theta \frac{2\mu_{1}\alpha_{1}w}{3L}\left[{\left(\frac{t}{2}+a\right)}^{3}-{\left(\frac{t}{2}\right)}^{3}\right]$$

*M_L_* is the bending moment of the bottom stiffness-compensating layer. We used a SUS304 stainless steel plate as the stiffness-compensating layer in this study. The bending moment *M_L_* is defined by:
(7)$$M_{L}=\theta \frac{E_{L}\left(w+2a\right)t^{3}}{12\left(1-\nu^{3}\right)L}$$
where Young’s modulus *E_L_* = 194,020 MPa and Poisson’s ratio *v* = 0.3.

To sum up, the explicit linear form of the relationship between input pressure and the bending angle output can be represented by the following:(8)$$\theta =\frac{A}{B+C+D+E}P_{in}$$
such that:$$\begin{array}{l} A=w\left(ab+\frac{bt}{2}+\frac{b^{2}}{2}\right)+\frac{\pi r^{2}}{2}\left(a+b+\frac{t}{2}\right)+\frac{2r^{3}}{3}\\ B=\frac{\mu_{1}\alpha_{1}}{L}\left[\pi {\left(a+b+\frac{t}{2}\right)}^{2}\left(a^{2}+2ar\right)+\frac{4\pi }{3}\left(a+b+\frac{t}{2}\right)\left[{\left(a+r\right)}^{3}-r^{3}\right]+\frac{\pi }{4}\left[{\left(a+r\right)}^{4}-r^{4}\right]\right]\\ C=\frac{4\mu_{1}\alpha_{1}a}{3L}\left[{\left(\frac{t}{2}+a+b\right)}^{3}-{\left(\frac{t}{2}\right)}^{3}\right],D=\frac{2\mu_{1}\alpha_{1}w}{3L}\left[{\left(\frac{t}{2}+a\right)}^{3}-{\left(\frac{t}{2}\right)}^{3}\right],E=\frac{E\left(w+2a\right)t^{3}}{12\left(1-\nu^{3}\right)L} \end{array}$$

Evaluation Index 1: Bending Resistance *BR*

Previous studies demonstrated that different parameters of the traditional FRAs (e.g., wall width, inner chamber height and width, and actuator length) alter bending behavior. In this section, the bending resistance (*BR*) is presented to evaluate the difficulty of bending deformation, defined by the ratio of the bending moment of the elastomeric matrix and the stiffness-compensating layer over the pressure of the bending torque applied:(9)$$BR=\frac{M_{b}+M_{L}}{M_{P}}$$

The lower bending resistance (*BR*) value indicates that the actuator curls more easily with low pressure.

### 3.2. Bending Model in Constrained Space

Most ADL tasks depend on finger flexibility, which not only includes finger flexion: stretching motion/extension is equally important, especially for the digit joint hypertonia of the subjects. Hence, joint stiffness and extension angle are utilized to quantitatively depict the abovementioned scenarios. In this section, the human–machine coupling interaction factor (e.g., joint stiffness) is examined. As shown in Figure 4, it is obvious that the wearer’s finger muscular tone/joint stiffness impacts the bending motion when the SCPBAs are coupled with a mannequin finger. Then, two different scenarios are investigated, as follows.

#### 3.2.1. Constrained Space with Intrinsic/Voluntary Flexion Torque

For able-bodied (AB) subjects, their fingers tend to curl inward naturally and show an initial position (*θ*_0_) because the muscular tone displayed in finger flexors (e.g., flexor digitorum profunda) is larger than in finger extensors (e.g., extensor digitorum profunda) when there is no exerted involuntary flexion torque and only intrinsic flexion torque *M_V_* existing in extension movement, as shown in Figure 4a. Passive extension resistance at the joints, indicated as intrinsic joint stiffness (*K_V_*), would be created to resist opposing movement when the flexor muscles stretch from the initial position (*θ*_0_).

**Figure 4 biomimetics-09-00638-f004:**
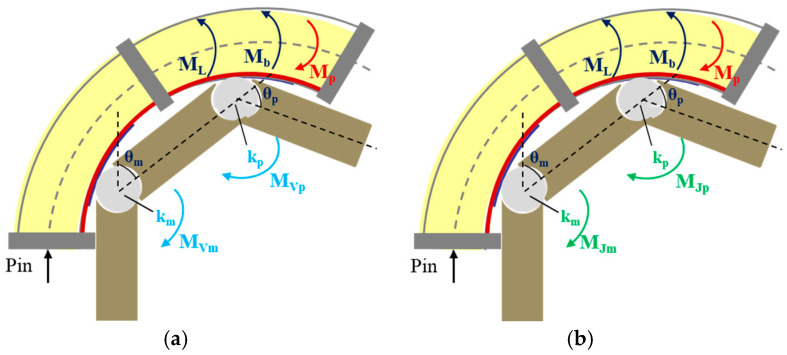
Diagram of bending deformation in constrained space considering finger joint stiffness. (**a**) Joints with intrinsic flexion torque; (**b**) Joints with involuntary flexion torque.

Assuming that the finger joint angle is consistent with the actuator–finger-coupled segmented bending angle (*θ*), the relationship between input pressure and coupled segmented angle can be redefined by the intrinsic flexion torque *M_V_*. Figure 4a depicts the moment equilibrium around the finger joints.

Case I: $0\leq \theta \leq \theta_{0}$ (intrinsic flexion torque *M_V_* exists in extension movement, and *M_P_* may be zero):(10)$$M_{P}+M_{V}=M_{b}+M_{L},M_{V}=K_{V}\left(\theta -\theta_{0}\right),M_{V}=M_{Vm}+M_{Vp}$$
(11)$$\theta =\frac{AP_{in}-K_{v}\theta_{0}}{B+C+D+E-K_{v}}$$

Case II: $\theta >\theta_{0}$ (intrinsic flexion torque *M*′*_V_* exists in flexion movement, and *M_P_* is not zero):(12)$$M_{P}=M_{b}+M_{L}+M\prime_{V},M\prime_{V}=K\prime_{V}\left(\theta -\theta_{0}\right),M\prime_{V}=M\prime_{Vm}+M\prime_{Vp}$$
(13)$$\theta =\frac{AP_{in}+K\prime_{v}\theta_{0}}{B+C+D+E+K\prime_{v}}$$

#### 3.2.2. Constrained Space with Involuntary Flexion Torque

For subjects with hand dysfunction (e.g., stroke survivors), strong extension resistance indicated as intrinsic joint stiffness (*K_J_*) would be induced due to the excess tone in finger flexors, called hypertonia. The involuntary flexion torque (*M_J_*) generated by extension resistance (*K_J_*) causes the finger to bend, or even form a hooked shape, as shown in Figure 4b. The following analysis quantifies the resistance due to hypertonia in terms of joint stiffness.

Case I: $0\leq \theta \leq \theta_{0}$ (intrinsic flexion torque *M_J_* exists in extension movement, and *M_P_* may be zero):(14)$$M_{P}+M_{J}=M_{b}+M_{L},M_{J}=K_{J}\left(\theta -\theta_{0}\right),M_{J}=M_{Jm}+M_{Jp}$$
(15)$$\theta =\frac{AP_{in}-K_{J}\theta_{0}}{B+C+D+E-K_{J}}$$

Case II: $\theta >\theta_{0}$ (intrinsic flexion torque *Mʹ_J_* exists in flexion movement, and *M_P_* is not zero):(16)$$M_{P}=M_{b}+M_{L}+M\prime_{J},M\prime_{J}=K\prime_{J}\left(\theta -\theta_{0}\right),M\prime_{J}=M\prime_{Jm}+M\prime_{Jp}$$
(17)$$\theta =\frac{AP_{in}+K\prime_{J}\theta_{0}}{B+C+D+E+K\prime_{J}}$$

To sum up, the intrinsic joint stiffness (*K_V_*) and the involuntary stiffness (*K_J_*) are presented in two different constrained scenarios. Since the major design consideration is for stroke survivors, only the joint stiffness upon extending the fingers is of interest (especially for the maximum extending angles (*θ*_0_
*− θ*, *θ = θ_smax_*, *M_P_* = 0)), and then the corresponding extension stiffness (*K_J_*) is the desired result. Further flexion of the fingers after the initial angle (*θ*_0_) is not researched in stiffness estimation. In the bending state of the SCPBAs, the cutoff pressure *P_in_* is defined as soon as the measured MCP angle exceeds its upper limit *θ*_0_, and therefore the SCPBAs are enabled no further.

Evaluation Index 2: Extension Angle *θ*_0_ − *θ*

The extension angle refers to the reverse bending of the finger caused by the stretching of the bottom rigid compensating layer when the finger is wearing the actuator. The difference between this angle and the initial angle (*θ*_0_) is the angle being sought.

Evaluation Index 3: Stiffness Evaluation *K_J_*

Case I: $0\leq \theta \leq \theta_{0}$ (intrinsic flexion torque *M_J_* exists in extension movement, and *M_P_* may be zero):(18)$$K_{J}=\frac{\left(B+C+D+E\right)\theta -AP_{in}}{\theta -\theta_{0}}$$

Case II: $\theta >\theta_{0}$ (intrinsic flexion torque *M*′*_J_* exists in flexion movement, and *M_P_* is not zero):(19)$${K^{’}}_{J}=\frac{AP_{in}-\left(B+C+D+E\right)\theta }{\theta -\theta_{0}}$$

In case I, the corresponding extension stiffness is of interest when the extension angle reaches its maximum (i.e., *θ*_0_ − *θ*, *θ* = *θ_smax_*, *M_P_* = 0).

## 4. Performance Characterization of the SCPBAs

### 4.1. Experiment Setup

To evaluate the bending performance and verify the analytical models, the proposed SCPBAs were clamped vertically upward to inflate repeatedly, reducing the influence of gravity. As shown in Figure 5, a filter regulator (MS4-LFR-1/4, Festo Inc., Esslingen, Germany) was used to filter and limit the maximum value of input pressure by manual operation in the inlet. The pressurized air regulated from the proportional solenoid valves (ITV2030, SMC, Tokyo, Japan) was continuously adjusted and then flowed into the SCPBAs and pressure sensors (SPTW-P6R-G14-VD-M12, Festo, Esslingen, Germany) monitoring the channels’ air pressure in real time. The SCPBAs’ bending angles were monitored by commercial bending sensors (FlexSensor 2.2/4.5, Spectrasymbol, Salt Lake City, UT, USA) with the stiffness-compensating layer attached to the bottom of the actuator, which was calibrated by three optical cameras (Optitrack, NaturalPoint, Corvallis, Germany) based on the relative position of the reflective markers. For the embedded bending path of the sensors, a DC–DC converter module (input: 8–40 V, output 5 V/3 A, VG10-T24033, DearRoad, Beijing, China) and a simple series divider circuit were connected, and then the voltage changes at both ends of the bending sensors were indirectly acquired in real time. Data from the cameras were streamed to a PC by Motive and synchronized with the pressure signal and voltages with a Beckhoff controller. The inflation pressure was applied in steps of 40 kPa until it reached 180 kPa without failure.

#### Characterization of the Embedded Flexible Bending Sensors

The embedded flexible bending sensors were calibrated to facilitate the ontological bending perception of the SCPBAs, which was assessed for the relationship between the electric signals (i.e., the voltage value) and the bending angles. The bending angles were indicated by the relative position of pasted reflective markers, and were captured using a 30 Hz OptiTrack system. For the sake of accuracy, the SCPBAs were actuated within a short interval of 20 kPa to retain data.

The bending angles of the sensors were captured by three cameras, calculated in Matlab 2018, and then plotted in Figure 6 (dotted line) with the corresponding voltages. The dotted line depicts the mean values of the real experimental measurements with the maximum and minimum errors, and the solid line indicates the fitted curve derived from experimental measurements. In Figure 6, the inset shows the fitted quadratic function model *Y* = 147.1068*x*^2^ − 947.1210*x* + 1528.2692 and the determination coefficient, *R*^2^ = 0.9792, of this sensor. The coefficient *R*^2^ was close to 100%, which indicated that the fitted model *Y* = *f*(*x*) matched the real scenarios, and then the bending kinematics were constructed in real time.

### 4.2. Bending Angle Measurement in Free Space

#### Changing the Stiffness-Compensating Layer Thickness

This section first investigates the proposed SCPBAs composed of hollow semi-obround elastomer with rigid rings in free space. The major parameters for the SCPBAs are (*a*, *b*, *r*, *w*, *L*) = (3.3, 7.5, 7.5, 15, 110). Four pieces of stiffness-compensating layers of 0.1, 0.2, 0.3, and 0.4 mm thickness, 15 mm width, and 60 mm (MCP segment) or 40 mm (PIP segment) length were compared in terms of their bending angles *θ_m_* and *θ_p_*.

As shown in Figure 7, all the soft joint angles maintained linear relationships with input pressure. Compared with the traditional FRAs without a stiffness-compensating layer (*t* = 0), the total bending angles versus the inflated pressure of the proposed SCPBAs were clearly lower. At 180 kPa, an increase in the thickness of the stiffness-compensating layer from 0.1 mm (50% of standard size, i.e., 0.2 mm) to 0.2 mm and 0.3 mm (150% of standard size) to 0.4 mm (200% of standard size) at the MCP and PIP segments decreased the magnitude of *θ_m_* from 61° to 59° and 55° to 39°, respectively, whereas the FRAs reached 67°. In addition, the magnitude of *θ_p_* decreased from 50° to 49° and 45° to 37°, and the FRAs reached 51°. Hence, with the increase in the thickness of the stiffness-compensating layer, the corresponding actuator allocated more energy to prompt stiffness-compensating layer bending, resulting in a reduction in total angle, as depicted by the bending resistance. Moreover, as the thickness of the stiffness-compensating layer increased continuously, the discrepancy between the modeling values and the real experimental measurements tended to decrease, which could be due to the dominant role of the rigid stiffness-compensating layer in the bending process, and the silicone rubber material had less impact on the changes.

Evaluation Index: Bending Resistance *BR*

In Figure 8, the lower bending resistance (*BR*) indicates that the actuator bends more easily with low pressure. Also, the thicker the stiffness compensating layer, the greater the corresponding bending resistance value. The maximum bending resistance value reached 63 when the 0.4 mm layer was integrated.

**Figure 7 biomimetics-09-00638-f007:**
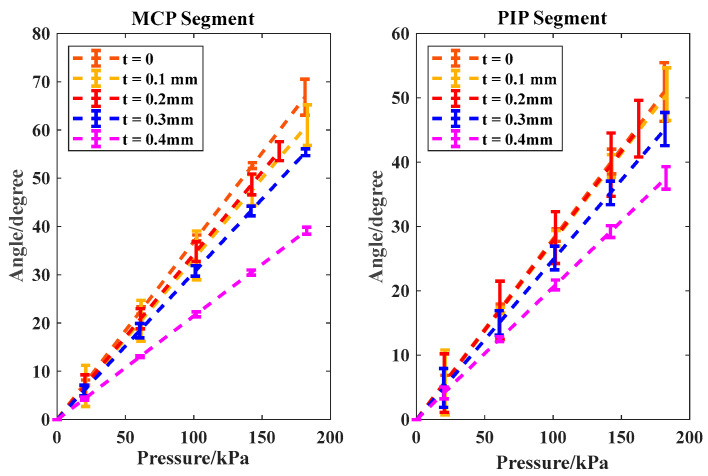
The relationship between the input pressure and output angle with various thicknesses of the stiffness-compensating layer in free space for MCP and PIP segments. The real experimental measurements are represented by all three trials’ error bars, and the dashed lines represent the modeling analytical results.

### 4.3. Bending Angle Measurement with Involuntary Flexion Torque

For subjects with hand dysfunction (e.g., stroke survivors), the modified Ashworth scale (MAS) is an assessment used in rehabilitation fields that relies on the extensive experience of a rehabilitation physician to obtain the muscle tone level of the hand. The relationship between MAS scores and joint stiffness magnitude has been researched [15,17,18,20], as summarized in Table 1. It is obvious that healthy subjects have lower joint stiffness with no voluntary flexion torque, so this scenario is identified as a special case of involuntary flexion torque (with hand dysfunction) to explore.

#### Extension Angle and Predicted Joint Stiffness on the Dummy Finger

Considering the thickness (*t* = 0.2 mm) and the deformation recovery property of the bottom stiffness-compensating layer, the torsional springs (*k*_1_ = 0.1321 Nm/rad and *k*_2_ = 0.1968 Nm/rad) were separately installed at the MCP joint position of the two 3D-printed index fingers, which were designed to mimic the affected fingers (Figure 9a). The SCPBAs with *t* = 0.2 mm are tested only on the MCP segment in this section, and the parameters of the torsion spring are indicated in Table A2. The theoretical stiffness of the torsional spring *k_T_* (Nm/rad) can be calculated using the following formula:(20)$$k_{T}=\frac{EI}{180Dn},I=\frac{\pi d^{4}}{64}$$
where *k_T_* is the stiffness coefficient of the torsion spring and *E* = 197,000 MPa is the elastic modulus of the spring material. *D*, *n*, and *d* indicate the mean diameter, number of windings, and the diameter of the spring wire for the torsional spring, respectively (Table A2).

For both various-stiffness dummy fingers, the trend of the analysis results was consistent. As shown in Figure 9b, a cutoff pressure of *P_in_* = 102 kPa was chosen for the upper limits of measured MCP joint angles and predicted stiffness. The actuation process of the SCPBAs was complete when the pressure was out of range. Since the experimental samples adopted a left-turning torsional spring, its stiffness coefficient decreased with the flexion direction in the extension phase. Hence, when the passive extension angle reached the maximum and the SCPBAs were not actuated (*θ* = *θ_s max_, M_P_* = 0), then the corresponding stiffness coefficient (*K_J_*) was the desired result. The estimated errors of the predicted stiffness were all less than 2.5%, which is far less than the 8% in Reference [18], and the minimum predicted error was 0.6% (the last column of Table 2). A maximum-extension bending angle of 22.3° was observed on the lower-stiffness finger at 0 kPa (the first two columns of Table 2). Although satisfactory results were demonstrated in slight spasticity with medium stiffness (0.04–0.5 Nm/rad), it was obvious that the estimated error was higher with low stiffness. This may have been because the elastic potential energy stored in the actuator was greater than the torsion spring when *P_in_* = 0 and the energy was mainly distributed on the PIP joint, resulting in larger estimation errors for the MCP joint.

## 5. Soft Robotic Glove with the SCPBAs for Task-Oriented Rehabilitation Training

The following experiments were conducted to verify the rehabilitation assistance of a soft hand exoskeleton system with the proposed SCPBAs in ADL tasks. Based on the resin support and tailored Velcro loop, a customized soft hand exoskeleton with the SCPBAs is displayed in Figure 10, which can achieve compatible motion and interaction with the hand safely and steadily. The lengths of the SCPBAs in the hand-wearable scenario are anthropomorphic. Three grasping scenarios in ADL tasks are demonstrated. Under an inflation pressure of 200 kPa, the soft exoskeleton can pick up objects with different postures, such as tip pinch.

### 5.1. Predefined ADL Tasks

Several ADL tasks have been designed to help improve hand function and conducted to verify the validity of hand exoskeleton systems [9,10,12,22,26]. Three hand-grasping types, power grasp (all five fingers), tripod pinch (thumb, index, and middle fingers), and tip pinch (thumb and index fingers), are frequently used during ADL tasks, helping subjects with impaired hand function to complete ADL tasks and improve their quality of life. Thus, in this study, three ADL tasks—power grasping an elliptical bottle (task 1), tripod pinching a medium-sized cuboid woodblock (task 2), and tip pinching a small cubic woodblock (task 3)—were tested, as shown in Figure 10. These objects are common in standard hand function tests, such as the box and block test (BBT).

### 5.2. Task-Oriented Rehabilitation Strategy

From the perspective of a professional rehabilitation physiatrist, a task-oriented rehabilitation framework was established. These three tasks were executed consecutively (task 1 to task 3). For a total rehabilitation trial, each task is required to be performed thrice, each lasting 20–22 s (5 s grasping object + 10 s holding object + 5–7 s releasing object). Thus, it takes 60–66 s to repeat the task three times, and one trial takes around 3.5 min (i.e., 3 tasks × 60/66 s per task). For a 1-h rehabilitation session, 18 repetitions (3.5 min × 18) are recommended, allowing for a rest of 2 min between trials to avoid muscle fatigue.

Enabled by the bending sensors, joint angles were detected, and the difference between the desired and actual bending angle were tracked in real time. Before starting the rehabilitation training, we first built a mapping relationship between the SCPBA bending angles and different tasks, taking the average of experimental results as desired targets. Table 3 demonstrates the targeted bending angles of the five fingers for three different tasks. The connective symbol (-) denotes that the changes in the finger for this task are ignored or the corresponding fingers are in free motion during training.

Figure 11 shows the bending angle and applied pressure as a function of time *t* (s) in the rehabilitation training trial for an able-bodied (AB) subject, which also represents the switch-mode rehabilitation from power grasp (task 1) to tripod pinch (task 2) to tip pinch (task 3). The solid blue curves represent the actual tracking bending angle of the involved fingers and the red curves show the corresponding targeted bending angles in Figure 11a. Figure 11b depicts the internally applied pressure for achieving the targeted angles.

The AB subject followed the experimental procedure and performed the task-oriented training mode. As depicted in Figure 11, the training commenced with a hand-relaxing phase, transitioning to a hand-closing phase. During the hand-closing phase, which lasted for 5 s, the target angle increase was triggered by a task-specific timer. The soft pneumatic glove then assisted in reaching the target angles during the hand-holding phase. For instance, the thumb achieved its target angle with a rapid response (within less than 2 s) and minimal overshoot (less than 5% of the target angle). Stability was maintained within 5 s, with the angle varying no more than 2% from the targeted angle. After 10 s, the target angle reduction commenced, and the hand-closing and hand-holding phases alternated. By adjusting the bending rate for each finger, the participant was able to achieve the required hand postures. One trial concluded with the sequential and repeated execution of three tasks.

However, Figure 11a highlights that the fitting between the desired angle and the actual angle worsened during the tip pinch task (task 3), particularly with the thumb angles. There are two reasons for this phenomenon: first, it may be related to the grasping posture, where the thumb and index finger adopt fingertip pinching, resulting in considerable instability during training. Second, the structure of the dorsal hand brace is also relevant. In this paper, the thumb actuator’s fixed position lacks degrees of freedom to coordinate flexibly with the index finger during task 3. In summary, these two possible reasons led to abnormal movement, especially for the thumb. In addition, the irregular shape of the object being grasped led to the middle and ring finger tracking performances worsening than the other three fingers during the power grasp tests (task 1).

Table 4 shows the tracking error in the rehabilitation training trial. The tracking errors in the training process can be calculated by the following formula:(21)$$Error=\frac{\theta_{t\_\mathrm{mean}}-\theta_{t\arg eted}}{\theta_{t\arg eted}},\theta_{t\_\mathrm{mean}}=\frac{1}{T_{task}}\cdot \sum_{t=1}^{T_{task}}\theta_{t}$$where *T_task_* is the task period of a finger’s targeted angle in the training process. For the thumb and index finger, *T_task_* is the period of task 1, task 2, and task 3. For the middle finger, Ttask is the period of task 1 and task 2. For the ring and little finger, *T_task_* is the period of task 1. *θ_t_mean_* is a finger’s mean values of measured angles in the corresponding finger’s period *T_task_*. *θ_targeted_* represents the targeted angles for the three ADL tasks mentioned in Table 3. Smaller errors indicate better tracking performance. For example, the tracking error of the middle finger in task 2 was as low as 5.93%, with an angle error of approximately 2°.

## 6. Conclusions

In conclusion, this paper presented the design, modeling, and evaluation of pneumatic bioinspired SCPBAs for hand rehabilitation in ADL tasks. The contributions of this research mainly include the following. (1) The SCPBAs take inspiration from the anatomy of human fingers, which makes the actuator match the finger profile and attain compatible motion coupled with a finger. (2) The proposed SCPBAs have significant advantages in the accuracy of predicted joint stiffness and extension angle. (3) The soft hand exoskeleton based on the SCPBAs integrated the proprioceptive stiffness-compensating layer, and was able to monitor the bending deformation and applied pressure in ADL tasks. Under an input pressure of 180 kPa, the proposed SCPBAs drove fingers to 113° with a maximum extension angle of 22.3°.

In future work, customized designs for multi-material three-dimensional technologies able to realize multi-gradient hardness will be utilized for stroke patients in hand rehabilitation training. Moreover, a feedback control system with flexible electronic sensing for wearable devices will be researched to enable more precise motion, accurately assess motor function, monitor a whole rehabilitation training trial, and achieve active training by users’ intention.

## Figures and Tables

**Figure 1 biomimetics-09-00638-f001:**
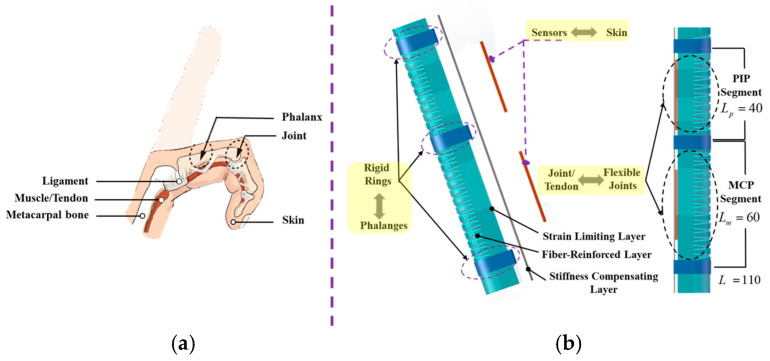
Schematics of the structure and concept of the proposed SCPBAs. (**a**) Illustration of the structure of a human finger with one flexion DOF. (**b**) Illustration of the flexible-joint–rigid-bone anatomy of a human finger.

**Figure 2 biomimetics-09-00638-f002:**
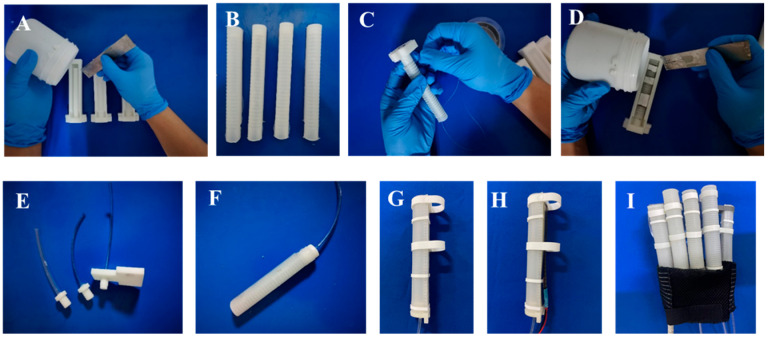
Fabrication and assembly of the SCPBAs.

**Figure 3 biomimetics-09-00638-f003:**
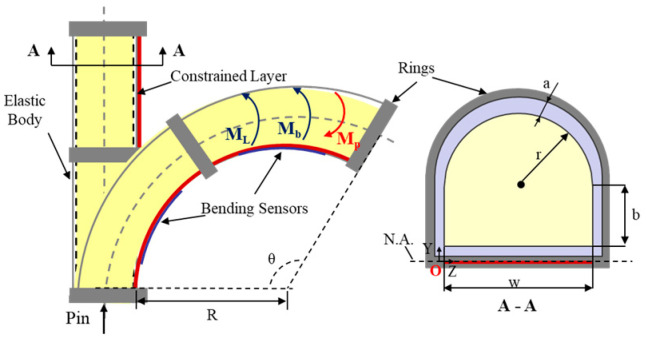
Illustration of the proposed SCPBA bending deformation in free space.

**Figure 5 biomimetics-09-00638-f005:**
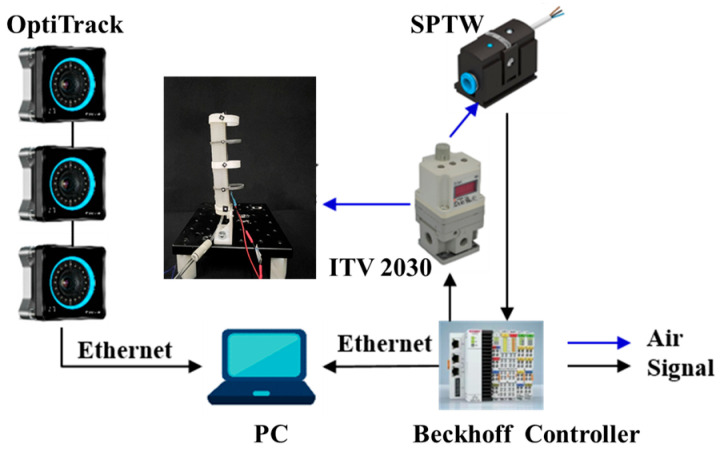
The experimental setup of the SCPBAs.

**Figure 6 biomimetics-09-00638-f006:**
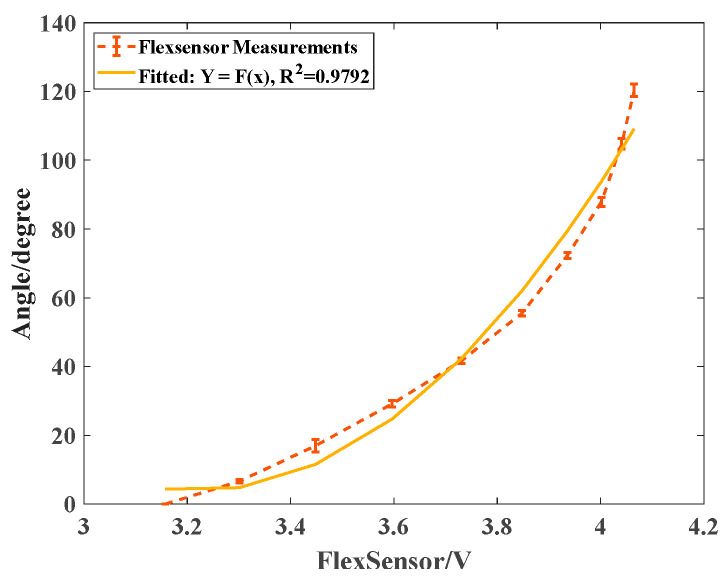
The relationship between voltage and corresponding angle of the FlexSensor attached to the SCPBAs.

**Figure 8 biomimetics-09-00638-f008:**
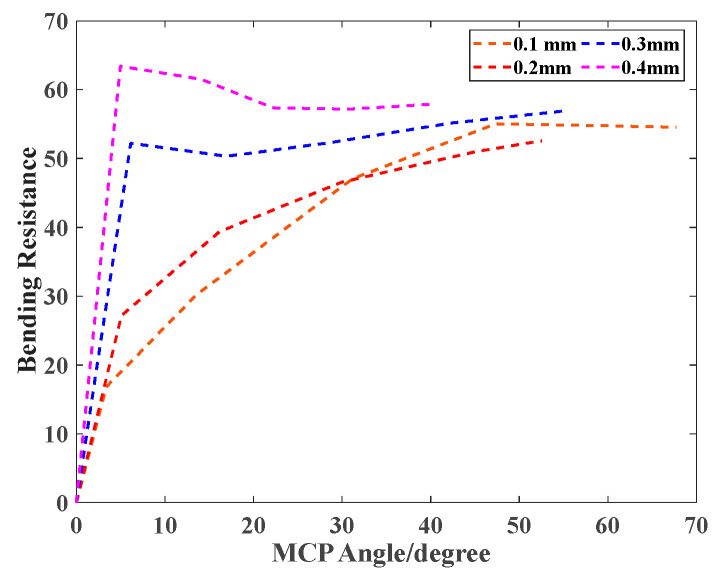
Bending resistance (*BR*)/efficiency comparison of the different thicknesses of the stiffness-compensating layer.

**Figure 9 biomimetics-09-00638-f009:**
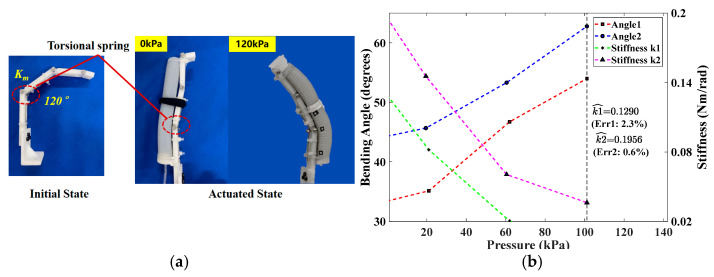
Stiffness estimation of the index finger MCP joint: (**a**) dummy finger coupled with torsional spring, (**b**) measured MCP joint angles and estimated stiffness.

**Figure 10 biomimetics-09-00638-f010:**
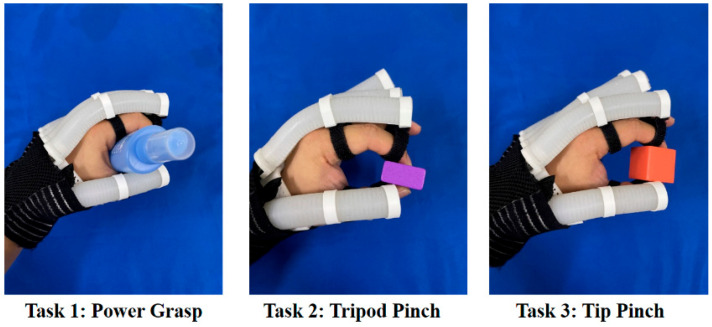
The three predefined ADL tasks.

**Figure 11 biomimetics-09-00638-f011:**
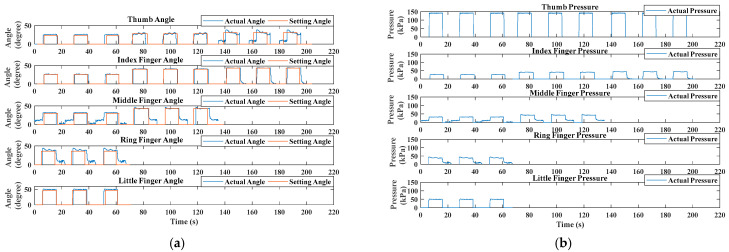
Experimental results of an able-bodied (AB) subject. (**a**) Bending angle results during the task-oriented training trial. (**b**) Applied pressure values for achieving targeted angles.

**Table 1 biomimetics-09-00638-t001:** Relationships between MAS scores and reference MCP joint stiffness [15,17,18,20].

Subject Condition	Stiffness Level	MAS Score	MCP Reference Stiffness (Nm/rad)
Healthy	Low Stiffness	0	≤0.04
Slight spasticity	Medium Stiffness	1+	$\left(0.04,0.5\right)$
Moderate-to-severe spasticity	High Stiffness	≥2	≥0.5

**Table 2 biomimetics-09-00638-t002:** The two evaluation indexes of the proposed SCPBAs for medium stiffness.

Stiffness Condition	Extension Angle (Degrees)		Stiffness Estimated (Nm/rad)		
	Ref. [10]	This Work	Reference Stiffness	Estimated Value	Error (%)
1+, Medium stiffness (0.04, 0.5), slight spasticity	[8.2°, 19.8°]	22.3	0.1321	0.1290	**2.3**
		16.8	0.1968	0.1956	**0.6**

**Table 3 biomimetics-09-00638-t003:** Experimental targeted angles for the three ADL tasks.

Tasks	Thumb	Index Finger	Middle Finger	Ring Finger	Little Finger
Task 1	23°	26°	30°	36°	47°
Task 2	28°	40°	43°	-	-
Task 3	31°	43°	-	-	-

**Table 4 biomimetics-09-00638-t004:** Tracking errors in the rehabilitation training process.

Tasks	Thumb	Index Finger	Middle Finger	Ring Finger	Little Finger
Task 1	12.63%	6.04%	6.07%	9.47%	15.90%
Task 2	9.35%	8.27%	5.93%	-	-
Task 3	8.79%	10.26%	-	-	-

## Data Availability

The data that support the findings of this study are available from the authors upon reasonable request.

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
