# Peer review of "A Pneumatic Soft Exoskeleton System Based on Segmented Composite Proprioceptive Bending Actuators for Hand Rehabilitation"

_biomimetics, 2024, doi:10.3390/biomimetics9100638_

Round 1

Reviewer 1 Report

Comments and Suggestions for Authors

The manuscript "A Soft Exoskeleton System based-on Segmented Composite Proprioceptive Bending Actuators for Hand Rehabilitation ADL Tasks" needs more clarity as well more details of the actuator itself. Also do not use abbreviation (ADL) in the title. Additionally the bending actuator being a pneumatic device need be given as well in title, otherwise such title do not match with content of manuscript. There are some other parts need to be better described as well some parts are missing

1. Please show the actuation mechanism in some scheme also hence authors refer to sensors how such movement sensed? Hence only air pressure included which parts give change of resistivity to get some sensing? Please give a better and more detail actuator description. Additionally please give more details from other authors where pneumatic muscles used for either robotics or sensing functionality in movements. Listing authors et al btw is not a good style in the introduction.

2. What part of the actuators are bioinspired? The movements of hands also are not really exoskeleton hence normal soft bending actuators based on hydrogels are much better as well having better sensibility. Please add more clarity in the manuscript. 

3. The actuator as pneumatic force how is the bending angle dependency on applied air pressure? Are long term measurements are made and the creep investigated? All this basic characterization before complex systems need to be investigated. The authors used in their modelling perfect behavior but how far is the modelling away from real measurements? How is the speed of actuation and how often can such "exoskeleton" used in a row? As shown in Figure 10 how different kind of person in hand shapes differ from each? Did the authors reproduce their results as shown in Figure 6 no mean values or standard deviation shown. Please give explanation for such.

4. In general knows from soft actuators used in smart gloves, the change of resistivity at bending are much lower as well those smart gloves can be operated by small batteries. In case of pneumatic principles how can that device used free? Please comment such.

5. There some minor parts in the text that need to be corrected

Page 1 line 44 " polygerinos .et" please correct that

Page 2 line 50 "et." there seems some missing

Page 3, line 132 "FRAs" never defined, please show what that abbreviation means?

Page 10 line 348 " hybrid actuator" what does such means, are different type of actuators combined?

Page 10 line 357 0.2mm or 0.2 mm, please check manuscript use always same with space or without

Comments on the Quality of English Language

Minor spell checking's are needed

Reviewer 2 Report

Comments and Suggestions for Authors

The paper describes the design and development of a soft glove for hand rehabilitation based on a novel bending pneumatic actuator. The device features a biomimetic design and an embedded sensor to real-time monitor the soft actuator bending angle. A mathematical model of the proposed actuator is also presented. The model and the prototype are validated through test-bench and experimental tests involving one subject. The bench tests verified the bending performance of the device and the effect of the construction parameters and the actuator-finger interaction on the device performance. On the other hand, the experimental tests involving the subject verified the device grip performance in a real context.

The bibliography appears recent and sufficiently exhaustive.

The research topic is interesting and has a prospect for real application, but some issues need to be addressed before publication:

1. The experimental test results discussion should be expanded. For example, Figure 11 highlights that the fitting between the desired angle (in red) and the actual angle (in blue) worsens during the tip pinch tests. In addition, during the power grasp tests, the middle and ring finger performances are worse than the other three fingers. Can the authors explain why?  

2. Table 4 shows the bending angle percentage errors. How are these errors calculated? Are tracking errors calculated over the whole test time? Or is it a maximum error between the desired and the actual angle? Or something else?

Minor comments:

1. P.2, line 61: MCP must be defined the first time it appears in the text.

2. P.2, line 70: DIP and PIP must be defined the first-time it appears in the text.

3. P.3, line 132: FRA must be defined the first-time it appears in the text.

4. Section 3.1: some equation parameters (MHC, MR, Mbottom, Mside, Mtop, t, τ, φ) should be better described. Maybe, some of them could be added even in Figure 4.

5. P.12, line 419: the text quotes reference 18 while Table 2 cites reference 10. Please check.

6. P.13, line 450: I guess the authors wanted to refer to Figure 10.

7. Figure 11 is barely readable.

8. Table 4 should be placed in the text after it has been mentioned.

Comments on the Quality of English Language

The writing process could be improved, I found some typos throughout the manuscript. 

Reviewer 3 Report

Comments and Suggestions for Authors

This paper presents a comprehensive design and evaluation of a soft hand exoskeleton prototype. The paper in general terms is well structured and the analytical computations are supported by the experimental results. The research subject is of clear interest since it targets an improvement in the quality of life of humans who have suffered a stroke. The bibliographic review presented also seems to be deep enough. Nevertheless, some issues must be corrected before publication:

Some acronyms are not properly placed. The first time MCP (line 61), DIP and PIP (line 70) are written, the reader does not know their meaning. They are explained later, in lines 121 and 122, but they should be explained the first time to reduce confusion. In a similar way, some acronyms (like SCPBA) are defined more than once (lines 20, 94, 158…), which is not necessary at all. Finally, some acronyms are never defined, as ANN, HPA, FRA...

When a reference is cited, sometimes there is a space before the reference number and sometimes not. The style should be coherent, so I recommend including a space always before any cite. In a similar way, when specifying the values of magnitudes sometimes there is a space between the value and the units and sometimes not.

In this reviewer’s opinion figure 1 is critical to understand the prototype functioning. Nevertheless, the graphical information provided is a bit confusing. Maybe figure 1 can be redesigned so the information was presented in a more clear way. Similarly, in figure 2 there is a collection of subfigures identified with different letters which are not explained. If that figure is included, it should be properly explained. The subfigures 4 (a) and (b) are quite similar, it should be helpful for the reader if authors highlight graphically the difference existing between them.

There is a general problem with the size of all figures, which is not compensated with the size of the text. All figures are too small and their size must be increased considerably. Please compare the size of the general text along the manuscript with the size of the text inside the figures.

The text size of some equations is also unbalanced, it is evident for example in eq(2) whose size is clearly too big. It can be also seen for example at line 207, where inside the same text line there is an undesirable mix of font sizes. The same happens with the numbers at lines 349 and 350. These are just some examples, the same issue occurs along the full manuscript, all sizes must be homogenized.

Maybe it is my fault, but I have not found the definition of the parameter ‘t’ appearing in some equations like eqs (4) to (7). Is it supposed to be a geometrical dimension like those ones appearing in figure 3?

Finally, at line 486 the figure 12 is cited and the paper has 11 figures.

Comments on the Quality of English Language

There are also some writing typos. I will highlight here some of them, but a careful review along the whole manuscript text needs to be done since there are more:

Line 32: stroke survivors of suffer -> often suffer?

Line 101: the improved analytical mode of the SCPBAs … are presented -> analytical models?

Line 180: to pursued the behavior -> to pursue?

Line 208: si can be obtain si=… ->obtained as

Line 208: coefficient l1, l2 and l3 -> coefficients

Round 2

Reviewer 1 Report

Comments and Suggestions for Authors

The authors gave good response to each point and improved the manuscript. Its now in much better form.